# Children in Need—Diagnostics, Epidemiology, Treatment and Outcome of Early Onset Anorexia Nervosa

**DOI:** 10.3390/nu11081932

**Published:** 2019-08-16

**Authors:** Beate Herpertz-Dahlmann, Brigitte Dahmen

**Affiliations:** Department of Child and Adolescent Psychiatry, Psychosomatics and Psychotherapy, University Hospital, RWTH, Neuenhofer Weg 21, D-52074 Aachen, Germany

**Keywords:** anorexia nervosa, childhood, children, review, comorbidity, medical assessment, treatment, outcome

## Abstract

Knowledge of anorexia nervosa (AN) in childhood is scarce. This review gives a state-of-the-art overview on the definition, classification, epidemiology and etiology of this serious disorder. The typical features of childhood AN in comparison to adolescent AN and avoidant restrictive eating disorder (ARFID) are described. Other important issues discussed in this article are somatic and psychiatric comorbidity, differential diagnoses and medical and psychological assessment of young patients with AN. Special problems in the medical and psychological treatment of AN in children are listed, although very few studies have investigated age-specific treatment strategies. The physical and mental outcomes of childhood AN appear to be worse than those of adolescent AN, although the causes for these outcomes are unclear. There is an urgent need for ongoing intensive research to reduce the consequences of this debilitating disorder of childhood and to help patients recover.

## 1. Introduction

Anorexia nervosa (AN) is a serious and often chronic disorder with a peak incidence in adolescence. Although childhood AN has long been recognized and described in several case studies [1,2], knowledge of the diagnosis and treatment has become increasingly important due to the rising admission rates of childhood AN in several European countries (see below). Although the same diagnostic criteria for AN should be applied to all age groups, there are differences in the presentation, epidemiology, medical and psychiatric comorbidities, as well as outcomes between younger and older patients.

Thus, the aim of this article is to give a thorough overview of the diagnosis, including behavioral and physical features of childhood AN, as well as of treatment and outcomes of the disorder. Moreover, we seek to underline the typical differences between onset in childhood and adolescence. Childhood will be defined as aged below 14 years, as this corresponds to the age span used in many investigations of young people with AN and will thus include premenarchal and postmenarchal onset of the disorder. In addition, age below 14 years is the legal age-definition of childhood in Germany.

## 2. Definition and Diagnosis

The diagnostic criteria in the classification systems “Diagnostic and Classification Manual of Mental Disorders – Fifth Edition” (DSM-5, APA 2013 [3]) and “International Classification of Diseases, Eleventh Revision” (ICD 11) [4] for childhood AN correspond to those in adolescence and adulthood. They focus on: A) restriction of energy intake with subsequent low weight, B) weight phobia and C) body image disturbance (DSM-5, [3]). The previous D criterion of DSM-IV [5], which focuses on amenorrhea, was left out in DSM-5 [3] to not preclude its applicability to premenarchal girls as well as to males and females on contraceptives. 

A significantly low body weight is defined as a weight that is less than minimally expected. However, what is the weight that is minimally expected? DSM-5 and ICD-11 [4] recommend using Body Mass Index (BMI)-for-age percentiles in children and adolescents for defining the weight threshold. While DSM-5 admits that a BMI below the 5th percentile may be too rigid for children in “light of failure of those children with a higher BMI-percentile to maintain their expected growth trajectory”, ICD-11 is stricter and lists a BMI beyond the 5th percentile as a required feature. This is difficult to understand as the weight criterion for adults (BMI of 18.5 kg/m²) corresponds to the 10th percentile; physical and psychological consequences might be even more severe in childhood or adolescent AN. Accordingly, most European and US-American studies use the 10th BMI-percentile as the weight threshold for a diagnosis of childhood or adolescent AN. 

Patients who fulfill all diagnostic criteria for AN with the exception of the weight criterion should be subsumed under the diagnostic category of “atypical AN”.

In ICD-11, two other essential alternative features are subsumed in the weight criterion section: rapid weight loss during the last six months, e.g., “more than 20% loss of total body weight” or failure to gain weight according to the individual developmental trajectory. According to previous studies [6,7] and our own clinical experience, rapid weight loss (weight loss in % per duration of illness) seems to be rather typical for children with AN. In those studies, patients had lost approximately 4% of their body weight/month in contrast to approximately 3% in adolescents [6,7].

## 3. Epidemiology

Generally, there are great differences in prevalence of AN in different regions of the world. Compared to Western countries such as Europe and the US, but also to China and Japan, there is a very low prevalence of AN in Latin America and Africa, but also among Hispanics in the USA [8]. Whereas the prevalence rate of AN in China amounts to 1.05%, it is below 0.01% in Africa. However, a general lack of representative epidemiologic data and different assessment methods limit the explanatory power of these statistical numbers [9]. Undoubtedly socio-cultural (and probably genetic) variety has an important impact on the development of AN in various areas of the world.

There is preliminary evidence that the age of onset of juvenile AN decreased during the last decade. Steinhausen and Jensen [10] reported that, in 2010, the most frequent age of onset ranged from 12 to 15 years, while the peak age of onset in 1995 was between 16 and 19 years. In the UK [11], the prevalence of AN in 10 to14-year-olds rose from 2.5/100,000 total children to 7.5/100,000 during the last two decades; the German health care statistics indicate an increase of patients below 15 years from seven to 13 from the year 2000 to the year 2017 (German Federal Statistics, http://www.gbe-bund.de/, assessed July 2019). Similar trends were observed in Italy and Portugal [12,13]. 

The exact prevalence and incidence of childhood AN are difficult to determine because most studies report joint data for children and adolescents. In the British surveillance study, the incidence estimate for AN according to the modified DSM-IV/ICD-10 criteria in patients ≤13 years was 1.09/100,000 total children of the same age. The youngest cases were detected between eight and nine years of age, with an increase in incidence until 12 to13 years [14]. The overall incidence estimate for eating disorders (EDs) in 12 to13-year-olds was 9.51/100,000 in the UK study, which was similar to the incidence of 9.4/100,000 in 10 to 12-year-olds of restrictive EDs in a Canadian pediatric practices’ surveillance study [15]. In a recent German registry study (recourse population), 25% of 289 patients who were admitted to inpatient care for the first time were below 14 years of age (childhood sample); 36.2% of the childhood sample was younger than 13 years [16].

Some studies have reported a higher percentage of boys in childhood in comparison to adolescent samples [7]. In the British surveillance study, the male to female ratio was approximately 1:9 (12% boys) [14] in comparison to 1:10 to20 in adolescence [17]. In the Canadian pediatric practices’ study of early onset restrictive EDs, this ratio amounted to 1:6 [15]. In the German registry study, the proportion of boys ≤14 years was very low (2.8%), similar to that in the adolescent sample [18]. In a recent Asian study, children treated for AN were also predominantly female (95.4%, [19]). This might point to the fact that male children with AN are likely grossly underdiagnosed and untreated. 

## 4. Symptomatology

As in adolescence, the disorder nearly always starts with dieting. At first, children often become vegetarians because they do not want to be involved in killing animals. They confine themselves to “healthy stuff,” such as vegetables, whole grain bread and fruit [20]. In contrast to girls with healthy leanness (BMI < 5th percentile), but no features of AN, they prefer food low in energy and fat [21].

Others are very active in sports, e.g., athletics, gymnastics or ballet. They compare their weight and shape with their peer sport’s group, and weight loss may become competitive. Other children cannot explain and do not seem to know why they started to control weight. They get preoccupied by thoughts of food and a morbid fear of fatness, need a long time to finish their meal and often take only very small bites. Food avoidance is observed in nearly 97% of children with restrictive EDs, preoccupation with food in more than 80% and an extreme fear of getting “fat” in more than 70% [15]. Some patients even refuse food completely and deny drinking because of an overwhelming fear of gaining weight. The relationship between calories and body fat is often poorly understood [1]; some patients do not even dare to swallow their saliva because they believe that it might contribute to weight gain.

Although one of the core features of AN is body image disturbance, according to DSM-5 and ICD 11, body weight and shape concerns are less frequent in children. In the study by Pinhas et al. [15], only slightly more than half found their body to be larger than it was. They often do not overestimate their body size but want to stay “as thin” as they are [22,23]. Many of them are physically hyperactive and often practice sports obsessively (see below for more details). Children are less likely to suffer from the binge/purge type of AN [7,18,19].

Observations on the clinical characteristics of early onset patients at admission to treatment also seem to depend on the cultural environment and health care system of the respective country. In two studies comparing patients with childhood and adolescent AN, children from Australia and Singapore showed a longer duration of illness and had lost weight more rapidly than adolescents [6,19]. Peebles et al. [7] observed a shorter duration of illness in children vs. adolescents in the US, and weight loss resulted in a significantly lower percentage of ideal body weight, which was not the case in the studies by Walkers et al. [6] and Kwok et al. [19]. In our own registry study, children had a shorter duration of illness and a higher BMI percentile at admission [18]. This was probably due to a higher recent awareness of pediatricians and parents to childhood AN in Germany because of detailed information about this disorder in the media. Moreover, in 1998, a regular health check-up for 12- to 15-year-olds (J1) was established in Germany, which covers a complete physical examination, including weight determination.

## 5. Medical Assessment

### 5.1. Symptoms and Complications

As children have a lower percentage of body fat than adolescents, the consequences of the same amount of weight loss might be more severe than in adolescents. A comprehensive history should be obtained and a comprehensive physical examination performed.

The typical clinical characteristics and complications of childhood AN are described in Table 1.

### 5.2. Endocrine Changes 

Starvation-induced hormonal disturbances are one of the major causes of medium- and long-term adverse effects of AN. The endocrine organs affected by AN are depicted in Figure 1. 

Due to immaturity of the child’s body and brain, endocrine disturbances may be even more devastating in childhood than in adolescence. The main endocrine changes are listed in Table 2.

### 5.3. Growth and Body Height 

Onset of AN might interrupt normal physical development, such as growth spurts and brain maturation [27]. Pinhas et al. [15] observed that nearly half of the patients presenting for ED under the age of 13 years had not grown during the last six months. In a large study including 211 adolescents with AN, patients’ height standard deviation scores at admission and discharge were significantly lower than in normal adolescents [28]. There is evidence that patients with childhood AN or onset of the disorder less than one year after menarche are at greater risk for growth retardation than those with later onset because the effects of starvation may interfere with the pubertal growth spurt (e.g., reduced insulin-like growth factor, type 1 (IGF-1)) [28]. In girls with premenarchal onset, catch-up growth is possible. However, weight gain is an essential precondition and must start in an age-limited time window before the ability to grow recedes. Once weight gain starts, the full effect on stature takes several years to evolve [29]. Growth is also disturbed in boys with AN. Those who recovered from their illness before the onset of their pubertal growth spurt had a significantly better catch-up growth curve than those whose disorder started after the beginning of the growth spurt [30].

Consistent with other studies (e.g., [31]), our own follow-up study of childhood AN also demonstrated that height at admission as well as at the five to ten-year follow-up was less than that of age-matched healthy adolescents. In our sample, at both assessment times, girls were on average approximately two cm smaller than the normal population. Thus, catch-up growth had not taken place. However, at first presentation, patients had already been ill for eight months on average, which might have contributed to the poor growth prognosis [32].

In sum, early diagnosis and treatment with a special emphasis on weight recovery might be the only way to prevent a disturbance of physical development.

### 5.4. Amenorrhea and Resumption of Menses

In our five to ten-year follow-up study of children hospitalized for AN (see above), only 60% resumed menses [32]. In a large multicenter study, patients with premenarchal onset of AN were at particular risk for protracted amenorrhea despite weight rehabilitation [33]. 

### 5.5. Impact on the Brain

In a systematic literature review on brain volume changes in AN, adolescents had a higher brain volume loss in gray and white matter than adults [34]. In addition to malnutrition itself, the starvation-induced deficits in sex steroid levels contribute to a disturbance in brain development [35]. Unfortunately, there are not enough longitudinal studies of adolescent patients to evaluate whether “scars” remain in the brain after recovery. To our knowledge, such studies in children do not exist. Thus, no clear conclusions can be drawn about the long-term consequences of brain volume loss in younger patients. However, it is highly probable that chronic, long-lasting AN inhibits normal growth of a developing brain and is potentially responsible for the neuroprogressive changes and neuropsychological deficits that are known to occur in patients with chronic AN [34].

## 6. Psychological Assessment

Assessment tools should not be used without a clinical interview of the parents and child by an experienced clinician. 

### Interviews and Questionnaires

Because instruments for adults and adolescents may not be psychometrically sound for children, a children’s version (ChEDE) of the Eating Disorder Examination (EDE; [36]), a standardized, semi-structured interview, was developed by Bryant-Waugh [37]. Findings from studies about the diagnostic quality of the ChEDE have been mixed; some authors complain about the low sensitivity of this interview in young populations, with the consequence of underdiagnosing ED (for more detailed information see Reference [38]).

The child version of a self-report questionnaire based on the Eating Disorder Examination Questionnaire (EDE-Q, [39]), the ChEDE-Q, was developed for children aged eight to 14 years [40]. It consists of 22 items, which are assigned to four subscales referring to the core symptoms of EDs: restraint, eating concern, weight concern and shape concern. The single scores may be added to one global score of eating disorder psychopathology. Additional key items measure diagnostically relevant symptoms of EDs, e.g., binge eating, loss of control, laxative or diuretics abuse, self-induced vomiting and fasting between binge attacks. This self-reported instrument was also evaluated in German children aged eight to 13 years by Hilbert et al. [41], with good psychometric properties.

Recently, a short form of the ChEDE-Q with only eight items was developed, which seems to be especially suitable for young children who are not able to complete long questionnaires [42].

Other self-report questionnaires include the Children’s Eating Attitudes Test [43] and the Children’s Eating Disorder Inventory [44].

## 7. Differential Diagnosis

### Medical Diagnoses

Important medical differential diagnoses are listed in Table 3.

Recently, an association between AN and autoimmune diseases, especially inflammatory bowel disease, celiac disease and diabetes mellitus type I, has become an emerging field of research. In a Swedish population-based nation-wide study, a bidirectional relationship was found between AN and celiac disease as well as between AN and Crohn’s disease, e.g., a diagnosis of AN increased the risk of later suffering from one of these autoimmune disorders and vice versa [45]. 

As a consequence, the German S3-guidelines for the diagnosis and treatment of EDs recommend excluding a diagnosis of inflammatory bowel disease (M. Crohn) by measuring calprotectin in stool, which is present in large quantities in the case of inflammation, and excluding a diagnosis of celiac disease by determining Tissue Transglutaminase IgA antibody and IgA antibodies, the latter of which is done to ensure that the patient produces enough IgA antibodies to render the analysis accurate (AWMF 2018, www.awmf.org/leitlinien/detail/ll/051-026.html).

## 8. Childhood AN and Avoidant/Restrictive Food Intake Disorder (ARFID) 

In a Swedish retrospective study, 58 patients who presented for treatment of AN under 13 years of age were compared to 44 patients suffering from low weight food intake disorder without any weight or shape concerns (definition corresponds to ARFID). Patients in the low weight food intake disorder group were younger at presentation and had a longer duration of illness, a lower maximum premorbid weight and a shorter stature. Thus, the physical consequences of low weight food intake disorder (ARFID) were even more severe than those of childhood AN, although patients with childhood AN received more intensive psychiatric care [46]. Interestingly, psychiatric comorbidity did not differ significantly between the groups: a prevalence of 48% was found in the AN group and of 41% in the low weight food intake disorder sample. Anxiety disorders and obsessive–compulsive disorder (OCD) were the predominant comorbid diagnoses in both disorders; however, depression was more frequent in the AN group [46].

## 9. Psychiatric Comorbidity 

In general, depression and dysthymia, anxiety disorders and obsessive-compulsive disorder (OCD) are the most frequent comorbid mental disorders in AN. This is also true for childhood AN [15,19,46]. Studies comparing the overall prevalence rates of psychiatric comorbid disorders could not identify any differences between childhood and adolescent AN [7,19]. However, Kwok et al. observed a large preponderance of OCD in childhood versus adolescent AN. According to our own experience, OCD is a frequent comorbid disorder in both age groups, with either washing or ordering compulsions or the obsession that things are going wrong [24]. Personality traits, such as perfectionism and scrupulosity, are also prominent in childhood AN. An association between AN and autism spectrum disorder is under intensive discussion. Children and adolescents with high functioning autism show an elevated rate of AN [47]. The exact prevalence rates of autism spectrum disorder in patients with childhood onset AN are not known. 

## 10. Etiology

### 10.1. Biological Risk Factors

There is no solid understanding for why some patients get ill during adolescence and others during childhood. In recent years, a genetic risk factor for AN and other EDs has become evident. Five to ten percent of female relatives of patients with AN also suffer from an ED. A recent genome-wide association study (GWAS) in nearly 17,000 patients with AN and 55,000 controls identified eight independent significant genetic loci [48]. Several important genetic correlations were found between AN and psychiatric disorders and AN and certain metabolic traits. In detail, there were significant correlations with OCD, major depressive disorder (MDD) (but not depressive symptoms), schizophrenia and anxiety disorders, but there were also significant inverse correlations with insulin metabolism (HOMA-IR(Homeostatic Model Assessment of Insulin Resistance)-Index, fasting insulin), leptin, body fat percentage and BMI. The authors concluded that, in the future, AN might be regarded as a psychiatric and metabolic disorder. 

There are some indications that early onset of a psychiatric disorder may be linked to a higher genetic loading, such as in schizophrenia or bipolar disorder [49,50]. A recent longitudinal study (Avon Longitudinal Study of Parents and Children (*n* = 1502, ALSPAC)) pointed to the fact that a low premorbid BMI in patients who later suffer from AN is present early in life. For male individuals, the average BMI growth trajectory has already diverged from the normal age-adapted population by age two; for females, this occurs before age four [51]. The authors point out that a low BMI in early childhood could indicate a “key biological risk factor”. A similar result was obtained in another large population-based study in Sweden. Women who weighed less than the reference group at birth were more likely to later develop AN [52]. However, other studies observed a rather high [53] or broad premorbid weight range [54], although these samples sizes were much smaller than those in the ALSPAC or Swedish study.

### 10.2. Sociocultural Risk Factors

In general, weight concerns or body dissatisfaction as expressions of a cultural slimness ideal emerged as risk factors for a variety of EDs. In retrospective studies, peer and family pressure were significantly associated with the development of AN and Bulimia nervosa (BN) [55]. It remains unclear at what ages and how sociocultural pressures affect children’s eating behaviors. 

Nearly 30% of 12-year-old schoolgirls reported that they compare their body to that of peers; approximately 12% admitted frequent comparisons [56]. In an earlier investigation, 22% of 12-year-old girls in a community sample wished to have a future BMI below the 10th BMI percentile [57]. In an earlier US-American study, 50% of 62 8 to 13-year-olds wanted to weigh less; 16% reported attempting weight loss. The vast majority of these young children already had a clear concept of dieting (either changing food choices or eating less food) [58].

Numerous studies have shown an association between media exposure and changes in body image. In a study by Götz and Mendel [59], the percentage of girls that regularly watch “Germany’s next top model”, a television show on adolescent fashion models, rose from 25% at age 9 to 65% at age 10. Of those who regularly watch “Germany’s next top model,” 64% consider themselves to be fat compared to only 41% who never watch the program. 

New (social) media are also contributing to the “thin role model” [60,61]. There are numerous pro-anorexia websites (pro-ana sites) that encourage dieting and also give advice on excessive exercise and purging. In a study on 1291 young adult pro-ED website users, usage of pro-ED websites was a significant positive predictor of EDE-Q scores [62]. However, there are no specific data about the influence of the internet on the development of EDs in children, although, in the light of increasing social media use by children, research in this field seems important [63]. 

## 11. Treatment

According to our own experience, children are much more difficult to treat than adolescents because of poor insight into their disorder. Children are often not able to express abstract concepts, such as motivation, self-awareness or body image concerns [64].

### 11.1. Medical Treatment 

Although outpatient treatment should be the first line treatment, some children might be medically unstable and will require inpatient treatment. A multimodal approach with a “developmentally aware and sensitive interdisciplinary team staff” experienced in the treatment of AN is the treatment of choice [65]. Depending on the health care system, child and adolescent psychiatrists and/or pediatricians, dieticians, physiotherapists and experienced nurses compose the core professionals [66]. Moreover, the British National Institute for Health and Care Excellence (NICE) guidelines [67] recommend educational and other age-appropriate activities to make reintegration to everyday life easier. This applies especially to children. 

During inpatient treatment, in settings without rooming-in of the parents many children feel homesick. If possible, children should stay at home for the weekend and see their parents as often as possible.

Nutritional rehabilitation is one of the most important goals of inpatient treatment. A meal plan consisting of six meals (three main meals, three snacks) should be established. Patients who are not able to eat should be supplied with liquid high-energy meals or with nasogastric tube-feeding. Detailed information on nutritional rehabilitation, which is quite similar to that of adolescent AN, can be found elsewhere [24,66]. There is no consensus about the definition of a healthy body weight and thus target weight. On the basis of previous studies, an age-adapted BMI percentile of 20 to25 is recommended as a reference weight to menstruate regularly [33,68]. However, there is growing evidence that the premorbid BMI, which may represent an individual’s normal growth track unaffected by the ED, might be a better target weight than a standardized BMI-percentile (see also References [31,51]). This might be especially true for children with atypical AN and a higher premorbid BMI.

### 11.2. Medication

There are few meta-analyses about the use of medication, especially neuroleptics and serotonin reuptake inhibitors, in adult patients with AN (e.g., Reference [69]), which could not show any effect on weight gain, eating disorder or general psychopathology. In a small RCT including adolescents from the age of 12 years onwards and young adults, no profit from olanzapine, an atypical neuroleptic, could be observed on weight gain [70]. 

To date, no medication for adolescent or childhood AN has been approved by the US Food and Drug Administration (FDA) or the European Medicines Agency (EMA). However, in a highly anxious or agitated state, the use of neuroleptics (e.g., olanzapine or risperidone) might be inevitable for a short time. 

### 11.3. Hormone Replacement Therapy 

The NICE guidelines [67] suggest considering a bone mineral density scan after one year of being underweight in children and adolescents. Because there is only a small time window for estrogen replacement therapy of children and adolescents with AN, and thus prevention of osteopenia and osteoporosis later in life, the NICE guidelines, German and Australian/New Zealand guidelines recommend considering incremental physiologic transdermal estrogen doses in girls with a bone age below 15 years to mimic pubertal estrogen increases [67,71] (AWMF 2018, www.awmf.org/leitlinien/detail/ll/051-026.html). This recommendation is based on an investigation by Misra et al. [72] who found an increase in bone mineral density in the spine and hip after transdermal estrogen application. 

### 11.4. Psychological Treatments

There are practically no investigations about the effects of treatment of AN in childhood. Several randomized trials have included children from 12 years on into their samples, such as those on family based treatment (FBT, e.g., Reference [73]), parent-focused treatment (PFT, [74]), systemic family therapy (SyFT) [75] and enhanced cognitive-behavioral psychotherapy (CBT-E) (from 11 years on [76]). However, in these studies, there were no evaluations of the impact of age; in other words, we do not know whether younger patients benefit similarly to adolescents. There is only one case series by Lock et al. [77], which comprises 29 children twelve years or younger with AN and three children with eating disorder not otherwise specified (EDNOS), restrictive type. During the course of treatment, children showed a statistically and clinically significant weight gain and decrease of eating disorder psychopathology. There were no significant differences in comparison to adolescents. 

Research on treatment setting is also scarce. There is one recent Cochrane analysis that compared the treatment of AN in an alternate setting (outpatient, day patient) to a specialist eating disorder inpatient program. Again, children from 12 years onwards were included. There was little or no difference in weight gain 12 months after the start of treatment between specialist inpatient care and active outpatient or combined brief hospital and day or outpatient care [78]. However, there was no evaluation of the impact of age. In our own study (included as one of the four trials in this Cochrane analysis), younger age in contrast to older age (adolescents) was an indicator of a worse outcome [79,80]. 

Parents have to be intensively involved in the treatment of children with AN. Thus, treatment strategies with a strong role for parents, such as FBT and PFT, seem to be highly suitable for young children. An educational group for parents may be an adjunct to other forms of treatment and may improve the parents’ knowledge, understanding and confidence, especially in early stages of the illness [81,82]. Parents often feel more relieved and understood if they can exchange experiences with other parents. However, in view of the fact that the outcome of childhood AN seems to be worse than that of adolescent AN (see below), there is an urgent need for ongoing research into effective treatment strategies in this age group (see below). 

## 12. Outcome

Studies on outcome in childhood AN are rare. Most investigations were conducted in the 1980s or early 1990s [83,84,85] and are based on inpatients admitted for AN. The duration of follow-up in these studies varied between five and nine years. Most of these studies suffer from methodological limitations, such as a large difference in the duration of follow-up within the single study and small sample sizes. The study by Higgs et al. [84] also included adolescents up to 16 years. To evaluate outcomes, most studies applied the Morgan and Russell outcome score (good outcome: weight has been maintained within 15% of average body weight, regular menses; intermediate: weight is only intermittently within 15% of average body weight and/or irregular menstruation; poor: weight continuously below 15% limit, absent or virtually absent menstruation). In these studies, 30% to 67% (on average 50%) of the sample achieved a good outcome, 7% to 35% an intermediate outcome (on average 25%) and 17% to 39% (on average 28%) a poor outcome. In contrast, follow-up studies in adolescent AN conducted at about the same time period reported a good outcome in up to 70% to 80% of the former patients [86,87]. 

In our more recent study, we investigated 52 subjects out of an original sample of 68 inpatients, with an average mean age of 12.5 years at admission to the hospital and 20.2 years of age at follow-up [32]. The patients were reinvestigated after an average time span of 7.5 years (range: 4.5 to 11.5 years). Approximately 40% of the former patients had a good outcome at follow-up, one third an intermediate and one fourth a poor outcome. One patient had died. 

In comparison, in a ten-year follow-up observation study of adolescents with AN based on a community sample, 50% had a good outcome and 25% a poor outcome [88]. In our own follow-up study in adolescent AN seven years after the first admission, 58% of the adolescents had a good outcome, 19% fell in the intermediate category and 22% had a poor outcome [89].

In the childhood onset study, there was a high comorbidity with other mental disorders, most prominently with affective, anxiety and substance-related disorders at the time of follow-up. The latter was rather surprising because, in previous investigations, comorbidity with substance abuse was not very common in restrictive EDs. There was also a high co-occurrence with personality disorders, especially avoidant and depressive personality disorders. Overall, 28% of the former patients fulfilled all diagnostic criteria for an additional psychiatric disorder at the time of reinvestigation. Most alarming was the poor mental health-related quality of life. Subjects with an intermediate or poor outcome of AN reported significantly reduced mental health. They scored even lower in a mental health survey than females with a chronic somatic and/or psychiatric disorder [32]. 

A recent Japanese study of 41 girls with restrictive AN and a long-term follow-up with an average length of 17 years observed more positive results: 71% of the former patients had achieved full remission, while 13% still suffered from an ED. Most participants had recovered after a time span of approximately 10 years, and some cases achieved remission even thereafter. Two patients died of suicide [90]. However, it is known from several follow-up studies that “time cures”: there are increasing recovery rates with progressing duration of follow-up [91].

Although the comparison between the follow-up studies mentioned above is not quite methodologically sound because of different periods of follow-up and large differences in the time-point of evaluation, there is some evidence that the outcome of childhood AN may be worse than that of adolescent AN. This was also confirmed by a recent long-term follow-up study in children and adolescents over a period of 30 years. Later age at onset predicted a better outcome, e.g., adolescent onset had a better prognosis than childhood onset based on the Morgan–Russell average outcome score [92]. 

In another recent long-term study outcome of childhood-onset, AN and low-weight ARFID were compared [93]. ARFID diagnoses were assigned retrospectively. At follow-up, approximately one fifth of the former patients with AN were still ill with an ED, and one fourth had another psychiatric diagnosis. No more diagnoses were assigned to approximately half of the patients. The ARFID group did not differ from the AN group with respect to the prevalence of EDs or other psychiatric diagnoses. However, all individuals in the ARFID group still fulfilled the diagnostic criteria of ARFID at follow-up, while the ED diagnoses in the AN group were more heterogeneous (mostly EDNOS). However, the results of this study probably have to be taken with care because there was a high attrition rate (45%). Nevertheless, according to these preliminary results, long-term follow-up in ARFID does not seem to be more favorable than that of childhood AN.

Follow-up studies of other mental diseases have confirmed that an early onset of a disorder might be associated with a worse course [94,95,96]. Genetic loading is likely higher in patients with early onset of a psychiatric disorder, or there might be more aversive early life events (or both).

## 13. Conclusions

In conclusion, there seems to be an increasing prevalence rate of childhood AN in many countries. Somatic and mental consequences of a long-standing ED in childhood are substantial and probably have a deleterious effect on later adult life. Regardless of the intervention, only approximately one-half of patients remain weight-restored in the long term. There are no special therapeutic strategies or settings for young children with AN, and little research has been undertaken to adapt treatment to younger patients. Thus, there is an urgent need for studies on strategies to achieve and permanently maintain remission and to help patients overcome this serious disorder of childhood.

## Figures and Tables

**Figure 1 nutrients-11-01932-f001:**
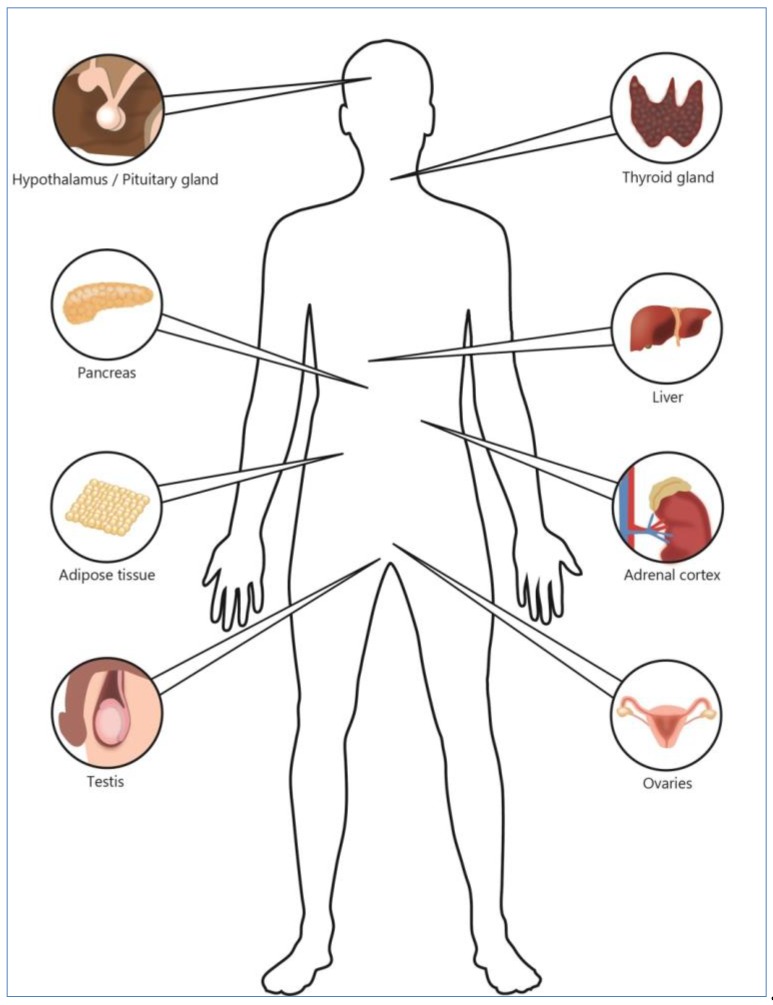
Endocrine organs affected by childhood AN.

**Table 1 nutrients-11-01932-t001:** Physical changes in childhood anorexia nervosa (AN) (after Reference [24]).

Organ System	AN
Physical examination findings	Dry skin
	Lanugo hair formation (only with severe weight loss)
	Jaundice (only with severe weight loss)
	Alopecia
	Brittle hair and nails
	Acrocyanosis
	Low body temperature
	**Dehydration**
	**Retardation of growth and pubertal development**
Cardiovascular system	Bradycardia (<50 bpm)
	Postural tachycardia (>20 bpm)
	ECG-abnormalities (mostly prolonged QT-interval, cardiac arrhythmia)
	Pericardial effusion (relatively frequent, but rarely dangerous)
	Heart murmur (mitral valve prolapse)
	Hypotension (<80/50 mm)
	Edema (before or during refeeding)
Gastrointestinal system	Impaired gastric emptying
	Reduced bowel sounds
	Constipation
	Pancreatitis
Blood	Leucopenia, thrombocytopenia, anemia
Biochemical abnormalities	Hypokalemia
	Hyponatremia
	Hypomagnesemia
	Hypocalcemia
	Hypophosphatemia (cave refeeding syndrome)
	Glucose ↓
	Creatinine ↑, urea nitrogen ↑
	AST, ALT (with severe fasting or beginning of refeeding)
	Amylase ↑, Lipase ↑
	Cholesterol ↑

Items in bold letters are frequently found in childhood AN. ↑ elevated; ↓ reduced. AST: aspartate aminotransferase; ALT: alanine aminotransferase (liver enzymes).

**Table 2 nutrients-11-01932-t002:** Endocrine changes in childhood AN (data from References [25,26]).

Hormone	AN
Thyroid axis	↓ fT3, n (↓) fT4
Gonadal axis (in postmenarchal childhood AN)	↓ FSH ↓ LH pulsatility ↓ Estrogens ↓ Androgens
Adrenal axis	↑ Cortisol n DHEAS
Growth hormone	GH resistance (↑ GH/↓ IGF-1)
Appetite-regulating hormones	↓ Leptin ↑ Ghrelin (fasting) ↑ (n) PYY (fasting)

↑ elevated; ↓ reduced; n normal; fT3 free triiodothyronine; fT4 free thyroxine; LH luteinizing hormone; FSH follicle-stimulating hormone; GH growth hormone; IGF-1 insulin-like growth factor, type 1.

**Table 3 nutrients-11-01932-t003:** Differential diagnosis of childhood AN [23].

Gastrointestinal Disorders
Inflammatory bowel disease
Celiac disease
Infectious diseases
Endocrine disorders
Diabetes mellitus
Hyperthyroidism (hypothyroidism)
Other endocrine disorders (e.g., hypopituitarism, Addison disease)
Other disorders
Central nervous system lesions (incl. malignancies)
Other malignant diseases
Superior mesenteric artery syndrome (more commonly a consequence of severe weight loss)

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
