# Peer review of "Children in Need—Diagnostics, Epidemiology, Treatment and Outcome of Early Onset Anorexia Nervosa"

_nutrients, 2019, doi:10.3390/nu11081932_

Round 1
Reviewer 1 Report
The present review has been well focused on relevant and interesting topic.
Authors well described the more frequent comorbid clinical conditions associated to Anorexia Nervosa. The section about psuchiatric comordifity sounds really good.
References are updated and well related to the main topic.
English is good.
No relevant concerns are present in the review.
Author Response
Dear reviewer,
I would like to thank you for your kind comments.
Kind regards
B. Herpertz-Dahlmann

Reviewer 2 Report
This review summarized children with AN, definition, diagnosis, epidemiology,symptoms, medical assessment as well as psychological assessment and treatment, thoroughly providing a full picture of children with AN. It is a great review article. One suggestion I have for the authors, could you include some information on what countries seem to be prevalent for AN, and the potential reason for that in sections like epidemiology? So we will know better the correlation between AN and countries or regions. Thanks for your attention.
Author Response
Dear reviewer,
I would like to thank your for your kind comments. In response, I have added a small paragraph in the epidemiology part of my article on the worldwide epidemiology of AN (although it is not quite the subject of this article), which is marked in yellow.
Kind regards
B. Herpertz-Dahlmann
